# Tear Film Stabilization and Symptom Improvement in Dry Eye Disease: The Role of Hyaluronic Acid and Trehalose Eyedrops versus Carmellose Sodium

**DOI:** 10.3390/jcm12206647

**Published:** 2023-10-20

**Authors:** José-María Sánchez-González, Carmen Silva-Viguera, María Carmen Sánchez-González, Raúl Capote-Puente, Concepción De-Hita-Cantalejo, Antonio Ballesteros-Sánchez, Lydia Ballesteros-Durán, Marta-C. García-Romera, Estanislao Gutiérrez-Sánchez

**Affiliations:** 1Department of Physics of Condensed Matter, Optics Area, Vision Sciences Research Group (CIVIUS), Pharmacy School, University of Seville, 41012 Seville, Spain; 2Department of Surgery, Ophthalmology Area, Medicine School, University of Seville, 41012 Seville, Spain

**Keywords:** dry eye disease, trehalose, hyaluronic acid, carmellose sodium, tear film stability, tear breakup time, non-invasive breakup time, standard patient evaluation of eye dryness, artificial tears, symptom management

## Abstract

This study evaluated the effectiveness of hyaluronic acid and trehalose (HA/trehalose) eyedrops in managing dry eye disease (DED) symptoms by measuring tear stability and administering a DED questionnaire. Sixty patients were treated with either HA/trehalose eyedrops (Tear A) or carmellose sodium eyedrops (Tear B) as controls. The tear breakup time (TBUT) and non-invasive breakup time (NIBUT) were monitored, and patients completed the standard patient evaluation of eye dryness (SPEED) questionnaire. After two months of twice-daily applications, patients treated with the HA/trehalose eyedrops demonstrated significant improvements in the NIBUT (12.98 ± 3.22 s) and TBUT (12.95 ± 2.98 s), indicating increased tear stability. Moreover, they reported lower dry eye sensation (6.70 ± 4.94 SPEED score points), suggesting a reduction in DED symptoms. These findings underscore the efficacy of HA/trehalose eyedrops in improving both the objective and subjective signs of DED, with twice-daily application enhancing ocular surface conditions and reducing patient-reported symptoms.

## 1. Introduction

Dry eye disease (DED) is a growing global health issue, identified via an array of symptoms that can severely affect the quality of daily life [1]. Multiple factors contribute to the development of this particular ocular disorder, including genetic predispositions, metabolic states, prolonged use of digital screens, the use of contact lenses or specific medications, and certain lifestyle choices [2]. As indicated in reports by the Tear Film and Ocular Surface Society (TFOS) and the Dry Eye Workshop (DEWS II) [3,4], hyperosmolarity and tear film instability emerge as the central causative factors in DED. These issues stem from a disruption in the homeostatic mechanism governing tear regulation, which, in turn, precipitates either a qualitative or quantitative deficiency in tears. This deficiency subsequently induces mechanical irritation on the ocular surface, moisture defects, tear film instability, increased friction, and hyperosmolar stress [5]. The culmination of these complications triggers a cascade of inflammatory responses and superficial damage, which correlate with the manifestation of several symptoms, such as photophobia, burning or itching sensations, blurred vision, a gritty feeling, and eye fatigue [6].

The quality of tear fluid is commonly gauged using tear breakup time (TBUT) [7,8]. Despite certain disadvantages associated with the use of fluorescein in assessing tear film stability, TBUT remains a prevalent method in DED management [7]. However, considering that the stability of the tear film could potentially be influenced by fluorescein, alternative measurement methods, such as the non-invasive breakup time (NIBUT), have been developed. Concurrently, dry eye symptoms can be quantified using the standard patient evaluation of eye dryness (SPEED) questionnaire, which has been proven to be a valid, reproducible, and reliable instrument [9].

As the incidence of DED continues to rise, a multitude of potential treatments are under investigation. Trehalose, a natural disaccharide composed of two glucose molecules linked via a 1–1 alpha bond, has been found to confer protection to epithelial cells [10], as well as a range of other cell types [11,12]. The cytoprotective properties of trehalose derive from its ability to safeguard cellular tissues and proteins against dehydration and oxidative stress-induced damage and denaturation [13]. Previous research on DED has demonstrated that trehalose reduces cell apoptosis and diminishes both oxidative and inflammatory activity at the ocular surface [14]. Studies have also explored the benefits of incorporating 3% trehalose as an adjuvant in standard treatment following laser-assisted in situ keratomileusis [15,16], as well as the efficacy and safety of artificial tear preparations containing trehalose and flaxseed oil in nano-emulsion forms [17], and the effects of novel in situ gelling artificial tear formulations containing lipids and trehalose [18].

Recently, a novel formulation combining hyaluronic acid and trehalose (HA/trehalose) in eyedrops was developed to leverage the bioprotective characteristics of trehalose [19,20,21,22,23,24]. This formulation, which is free of preservatives and available in a multi-dose preparation, facilitates prolonged use without inducing damage to the ocular surface. The main aim of this study was to assess the efficacy of HA/trehalose eyedrops in alleviating the symptoms of dry eye disease (DED). We focused on two key parameters: tear film stability, measured through fluorescein tear break-up time, and patient-reported symptoms, captured via a targeted DED questionnaire.

## 2. Materials and Methods

### 2.1. Design

This study, conducted between February 2021 and April 2021, was designed as a randomized, prospective, simple-blind, single-center trial. It took place within the facilities of the Optics and Optometry department at the School of Pharmacy, University of Seville, Spain. This study was meticulously designed and executed to conform to the highest scientific and ethical standards, guided by the principles enshrined in the Declaration of Helsinki and the ethical guidelines provided by the Ethical Committee Board of Andalusia.

### 2.2. Subjects

A total of 60 participants, comprising 29 males and 31 females, were enrolled in this study. Participants were stratified into two groups of 30 each. Group A, the treatment group, had an average participant age of 22.00 ± 0.95 years, with ages ranging from 21 to 26 years. Group B, the control group, had a similar demographic profile, with a mean age of 22.30 ± 2.20 years and ages ranging from 19 to 26 years. Each participant was thoroughly briefed about the study’s protocol and had any questions or concerns addressed prior to signing an informed consent form.

The study participants were selected based on specific inclusion and exclusion criteria. The inclusion criteria were as follows: (1) age between 18 and 30 years, (2) diagnosis of DED evidenced by symptoms such as photophobia, burning or itching, blurred vision, or gritty sensation, and (3) a non-invasive breakup time (NIBUT) of less than 15 s. The exclusion criteria were as follows: (1) the use of soft or rigid gas-permeable contact lenses, (2) the use of eyedrops in the month prior to the study, (3) any previous ocular surgeries, (4) inability to provide informed consent, and (5) inability to comply with the proposed follow-up schedule.

### 2.3. Materials and Instruments

Several tools and materials were used in this study to ensure the accurate measurement of variables. A slit lamp (TOPCON SL-6E, Tokyo, Japan) was employed for detailed observations of the tear film. To measure the tear breakup time (TBUT), fluorescein strips (Bio Glo ContaCare Ophthalmics & Diagnostics, Gujarat, India) impregnated with a saline solution were used. The upper bulbar conjunctiva was stained with the fluorescein strips after application, and the tear film was observed under cobalt blue illumination using the slit lamp. For the TBUT test, the slit lamp microscope was adjusted to specific settings to ensure accurate and consistent observations. The illumination system was set to a blue cobalt filter to enhance the visibility of the fluorescein dye. The beam width was adjusted to approximately 10 mm, and the height was set to cover the entire corneal surface. The magnification was dialed to either a 10X or 16X objective lens, depending on the level of detail required. The angle of the observation system was set at 45 degrees to the patient’s visual axis to enable a clear, unobstructed view of the tear film and corneal surface.

Under normal conditions, the tear film displayed a uniform fluorescein distribution, a stable break-up time of more than 10 s, and a smooth corneal surface. In contrast, dry eye disease (DED) conditions manifested a shorter break-up time (of less than 10 s), non-uniform fluorescein spreading, and signs of corneal surface disruption. These observed differences under the slit lamp are crucial for the accurate diagnosis and management of DED.

The NIBUT measurements were carried out with a Placido-disc-based topographic system (Topcon CA-200F Corneal Analyzer, Boisbriand, QC, Canada), which delivers accurate and high-resolution images of the anterior corneal surface. All patients completed the SPEED questionnaire for the assessment of DED symptoms. In our study, both the TBUT and the NIBUT were measured three times, and the average was taken for analysis.

Tear A eyedrops (Thealoz DUO^®^, Thea^®^, Clermont-Ferrand, France), containing 3 g of trehalose and 0.15 g of sodium hyaluronate in a 100 mL isotonic buffered solution, were used in this study. The control, tear B eyedrops (Xailing Fresh^®^, VISUfarma B.V^®^, Amsterdam, The Netherlands), contained 0.5 g of carmellose sodium in a 100 mL isotonic buffered solution. Both eyedrops were preservative-free lubricants and were packaged for daily use. We confirm that all the products that were used in this study were stored according to the manufacturer’s recommended storage conditions.

### 2.4. Procedure

This study was divided into three phases. The first phase involved identifying potential patients and evaluating their eligibility based on the inclusion and exclusion criteria. The second phase involved pre-treatment tear measurements following a month-long washout period, wherein patients were instructed to abstain from the use of artificial tears or eyedrops. Upon completion of this phase, patients were randomly assigned to either the treatment or control group and instructed on the correct technique for applying artificial tears. To ensure consistency in both treatment and evaluation, the subjects in our study received doses in both eyes. The posology for the treatment regimen entailed administering one drop per eye every 12 h. The final phase involved post-treatment tear measurements after a two-month period of eyedrop usage. To maintain blinding, the tear measurements were carried out without knowledge of the patient’s group allocation. Each patient was assigned a unique identifier to preserve anonymity.

### 2.5. Interim Assessments and Adverse Events Monitoring

In order to assess the treatment progression and observe any adverse events, interim assessments were conducted bi-weekly. Participants were questioned about any discomfort, redness, itching, or other adverse events that they may have experienced. They were encouraged to report any such occurrences immediately, even if they occurred outside of the scheduled assessments.

All adverse events, regardless of perceived relation to the treatment, were meticulously documented and reported to the ethics committee for further investigation. Any participant who experienced severe or intolerable discomfort was offered an immediate discontinuation of the study. However, no such events occurred during this study.

### 2.6. Follow-Up Procedures

After completion of the two-month treatment phase, all participants returned for a final assessment, which involved a post-treatment tear measurement. During this visit, participants were asked about their overall experience, including the severity of any adverse events, the tolerability of the eyedrops, and whether they perceived any improvements in their DED symptoms.

A follow-up period of one-month post-treatment was maintained, during which the participants were monitored for any persistent or delayed adverse effects. However, no complications were noted during this period. This continuous monitoring added an additional layer of safety and reliability to the results of this study.

### 2.7. Randomization, Allocation, and Blinding

Participants were randomly assigned to either the treatment or control group in a 1:1 ratio using a computer-generated randomization list. The randomization procedure was overseen by an independent statistician not involved in this study.

Allocation to the groups was concealed from both the participants and the investigators to prevent bias. To achieve this, a research assistant not involved in the study was responsible for the distribution of the eyedrops to the participants.

To maintain the simple-blind design, neither the participants nor the investigator who performed the tear measurements were aware of the group allocation. This methodological approach was designed to prevent any form of bias that could influence the outcome of this study.

### 2.8. Quality Assurance

In order to ensure the quality of the results, several measures were taken. First, the study protocol was rigorously followed, and any deviations were duly noted and reported. Second, all measurements were performed by experienced investigators who were regularly trained and supervised. Third, all the equipment used was regularly calibrated and maintained, in accordance with the manufacturer’s guidelines.

Any missing or doubtful data were carefully handled. Missing data were treated as lost completely at random, and an intention-to-treat analysis was performed to maintain the randomness of the allocation. Doubtful data were cross-checked and confirmed by a second investigator.

### 2.9. Data Management

The data collected were meticulously managed and safeguarded to maintain the confidentiality and anonymity of the participants. All data were stored securely in password-protected electronic databases. Only the primary investigators and authorized personnel had access to these databases. Backups of the data were regularly conducted to prevent data loss.

### 2.10. Statistical Analysis

All collected data were analyzed using SPSS 26.0 for Windows (SPSS Science, Chicago, IL, USA). The Shapiro–Wilk test was used to assess data normality. Descriptive analysis of the data was first conducted, followed by comparative analyses using the Student’s *t*-test for related samples for normally distributed data and the Wilcoxon signed-rank test for non-normally distributed data. The magnitude of the treatment effect was calculated using Cohen’s d formula, and all statistical tests were performed at a 95% confidence interval with the significance set at *p* < 0.05.

## 3. Results

At the baseline, group A’s mean age was 22.00 ± 0.94 (21–26) years old, while group B’s mean age was 22.48 ± 2.29 (19–26) years old, with no statistically significant differences observed (*p* = 0.33). In our study, the 60 participants were classified into three categories of DED severity based on their TBUT results, following the criteria established by the TFOS DEWS II [1,7]: mild (with a TBUT between 8 and 15 s), moderate (with a TBUT between 5 and 7 s), and severe (with a TBUT under 5 s). In Group A (*n* = 30), the distribution was as follows: twelve patients (40%) fell into the severe category, ten patients (33.3%) were classified as moderate, and eight patients (26.7%) were considered mild. In Group B (*n* = 30), the breakdown was as follows: ten patients (33.3%) were categorized as severe, fourteen patients (46.7%) as moderate, and six patients (20%) as mild. Regarding gender, group A had 14 males and 16 females, while group B had 15 males and 15 females. The average NIBUT value of group A was 5.65 ± 2.15 (1–10) seconds, while group B’s average NIBUT value was 6.03 ± 1.61 (3–11) seconds; however, the difference was not statistically significant (*p* = 0.27). The average TBUT result of group A was 4.53 ± 2.02 (1–13) seconds, while group B reported an average TBUT of 5.18 ± 1.53 (3–10) seconds, with no significant differences observed (*p* = 0.05). Finally, the average SPEED test result of group A was 8.80 ± 4.64 (2–20) score points, and an average of 9.90 ± 2.09 (6–15) score points was calculated for the carmellose control group, with no significant difference between these values (*p* = 0.24).

### 3.1. The NIBUT

The average non-invasive tear quality value of group A was 12.98 ± 3.22 (6–20) seconds (t = 15.89, *p* < 0.01), demonstrating a large effect of 2.67; the NIBUT increased by 7.33 ± 3.57 (6.41–8.25, 95% confidence interval) seconds. In contrast, the carmellose control group’s average value was 6.10 ± 1.77 (2–10) seconds (Student’s *t* = 0.51, *p* = 0.61), revealing an effect of 0.04; the NIBUT increased by 0.06 ± 1.00 (0.13–0.32, 95% confidence interval) seconds. These differences are shown in Figure 1. The post-treatment comparison between these groups demonstrated that group A had a better result, with a difference of 6.88 ± 0.47 (5.94–7.82, 95% confidence interval) seconds (*t* = 14.49, *p* < 0.01).

### 3.2. The TBUT

The invasive tear quality test results of post-treatment group A revealed an average TBUT value of 12.95 ± 2.98 (7–20) seconds, (*t* = 20.67, *p* < 0.01). This indicated a large effect of 3.30, with a TBUT increase of 8.41 ± 3.15 (7.60–9.23, 95% confidence interval) seconds. The post-treatment carmellose control group’s TBUT average value was 5.42 ± 1.70 (2–9) seconds (*t* = 1.65, *p* = 0.10), indicating an effect of 0.14; the TBUT increased 0.23 ± 1.09 (from 0.14 to 0.51, 95% confidence interval) seconds. These differences are shown in Figure 2. The post-treatment comparison between these groups demonstrated that group A achieved better results, with a difference of 7.53 ± 0.44 (from 6.65 to 8.41, 95% confidence interval) seconds (*t* = 16.97, *p* < 0.01).

### 3.3. SPEED Questionnaire

The average result from the SPEED questionnaire on patient perception of dry eye sensation in post-treatment group A was 6.70 ± 4.94 (from zero to twenty) score points (*t* = 2.81, *p* < 0.01). This outcome demonstrated a moderate effect of 0.44. The SPEED score decreased to 2.10 ± 4.08 (from 0.57 to 3.62, 95% confidence interval) points. The carmellose post-treatment control group had an average score of 9.53 ± 1.92 (from six to thirteen points, *t* = 0.84, *p* < 0.01) demonstrating a small effect of 0.18. The SPEED score decreased 0.36 ± 2.38 (from 0.52 to 1.25, 95% confidence interval) points. These differences are shown in Figure 3. The post-treatment comparison between these groups demonstrated that group A achieved better results, with a difference of 2.83 ± 0.97 (from 0.87 to 4.79, 95% confidence interval) score points (*t* = 2.92, *p* < 0.01).

## 4. Discussion

Dry eye disease (DED) represents a prevalent ocular condition linked to alterations in the tear film, requiring therapeutic strategies that aim for its restoration. This research endeavored to compare the efficacy of eyedrops containing a combination of hyaluronic acid and trehalose (referred to as HA/trehalose hereafter) against a formulation solely containing carmellose. The need for such a comparison arises from the ongoing advancement in the therapeutic approach towards DED, involving artificial tears with differing compositions. One of the latest additions to these formulations was trehalose. This disaccharide, known for its cytoprotective properties [10,13], has shown promise in the amelioration of DED symptoms. This potential was notably demonstrated in a recent study, where trehalose was found to effectively treat patients with moderate-to-severe DED [10].

In the present study, patients were divided into two groups: the HA/trehalose group (group A) and the control group. The primary goal was to evaluate the effects of these treatments on tear stability, as gauged via the TBUT (tear break up time) and the NIBUT (non-invasive break up time). Our findings demonstrated that the TBUT significantly increased by 8.41 ± 3.15 s in the HA/trehalose group. Interestingly, this clinical enhancement was not witnessed in the control group. Such results, showing the efficacy of HA/trehalose, align with previous work that examined a higher posology of trehalose eyedrops [22,23]. Interestingly, these studies concluded that the application frequency of artificial tears did not necessarily contribute to an increased effect on dry eyes beyond two times a day. However, a placebo effect on perceived symptomatology might have been present due to the subject’s belief of taking a higher dosage. The SPEED (standard patient evaluation of eye dryness) questionnaire scores [9] further reinforced the benefits of the HA/trehalose formulation. This validated patient-reported outcome measurement tool revealed a notable improvement in DED symptoms in the HA/trehalose group. There were some symptom improvements observed in the control group as well, albeit not statistically significant in our cohort.

Such results indicated that the overall satisfaction of patients in the treatment group led to a significantly higher appreciation score than in the control group. This is despite the fact that the SPEED questionnaire is not the most commonly used test; however, its effectiveness in DED assessment has been demonstrated in other studies [25,26]. Our findings, showing reduced subjective discomfort symptoms, less damage to the eye surface, and improved tear stability after treatment, are consistent with the results reported in earlier studies [17,20,21,22,24,27,28].

In the studies conducted by Cagini et al. [20,21], Mencucci et al. [24], and Karaca et al. [28], the effectiveness of hyaluronic acid (HA) combined with trehalose, in both ophthalmic solution and eye drop form, were evaluated in the context of post-operative ocular surface inflammation, post-operative discomfort, tear film stability, and dry eye symptoms. All their studies indicated that treatments involving HA and trehalose effectively ameliorated these ocular complications. Interestingly, in the two studies conducted by Cagini et al. [20,21], patients treated with trehalose and HA eye drops demonstrated a quicker recovery of the normal ocular surface parameters (including the TBUT, CFS, OSDI scores, and Schirmer test) compared to those receiving other treatments, including HA eye drops alone or no treatment at all. This suggests that trehalose may have an additive or synergistic effect with HA in speeding up ocular recovery and reducing inflammation post-surgery. Similarly, the study conducted by Mencucci et al. [24] found that the peri-operative use of a HA/trehalose ophthalmic solution reduced post-cataract surgery dry eye signs and symptoms in patients with mild/moderate dry eye disease. These effects were particularly pronounced when the solution was administered both pre- and post-operatively, underscoring the importance of preventative care in mitigating post-operative complications.

Contrastingly, the study published by Karaca et al. [28] showed that the ocular residence time of a 0.3% HA solution was longer than that of a solution containing trehalose, HA, and carbomer. This longer residency time also correlated with prolonged patient comfort. Although the overall tear meniscus and osmolarity parameters improved similarly with both lubricants, these findings suggest that certain factors, such as comfort duration, should also be considered when choosing an optimal treatment strategy for dry eye disease. It is also important to mention that none of the studies reported any severe adverse reactions or side effects to the treatments. This, in combination with the reported benefits, highlights the potential of HA/trehalose-based formulations as safe and effective treatment options for a variety of ocular surface disorders and post-operative complications. Nonetheless, further research is required to understand the specific mechanisms by which these treatments function. Such knowledge could help in optimizing these formulations and provide insight into whether their effectiveness can be improved with other adjunct therapies. Moreover, future studies should also explore the potential use of these treatments in other ophthalmological indications.

However, in contrast to these previous studies [20,24], where the HA/trehalose formulation was administered more frequently per day and for only one week, our study extended the treatment period. Furthermore, we observed a decrease in discomfort symptoms related to vision functionality and environmental triggers post-treatment. These findings strongly suggest that additional research is needed to explore the specific impacts of HA/trehalose eyedrops on perception mechanisms. This study results demonstrated statistically significant differences in the NIBUT, TBUT, and SPEED questionnaire scores, with the HA/trehalose group outperforming the carmellose group. This positive outcome could be attributed to the limited daily treatment frequency requirement in our study, possibly leading to improved patient compliance. Other large-scale studies have similarly confirmed that HA/trehalose eyedrops were significantly more effective than a saline and hydroxypropyl methylcellulose solution [10,27]. One particular study highlighted that HA/trehalose eyedrops effectively increased the tear film thickness for 240 min, as measured with a high-resolution optical coherence tomography system. This pointed towards a longer corneal residence time, signifying the sustained efficacy of this formulation [29].

Interestingly, another study, published by Orobia et al. [16], which investigated a treatment amalgamating carmellose and hyaluronic acid not only reported just an improvement in DED but also suggested its applicability across all DED severity degrees. However, a study by Diaz-Llopis et al. [30] demonstrated a more significant improvement in symptoms and a reduction in pro-inflammatory molecules in DED patients using seawater artificial tears than in those using a carmellose formulation. This indicates that carmellose, in isolation, may not bring about the desired improvement in dry eye conditions, prompting the need for it to be combined with another wetting agent.

### 4.1. Limitations

This study had several limitations that should be taken into account when interpreting these results. The sample size was relatively small (60 patients), and there was a bias towards individuals with a specific severity of DED, which may thereby limit the generalizability of these findings. The treatment period was restricted to two months due to feasibility issues; hence, longer follow-ups are needed to confirm these observed outcomes.

Moreover, patients were not blinded to the assigned treatments, introducing the potential for subjective bias. The tests that were used to measure the TBUT and NIBUT carry an element of subjectivity, due to their reliance on human observation, which may introduce a margin of error. Finally, this study did not control for the exact amount of the dose administered to each patient, which may have influenced the treatment’s efficacy and repeatability.

### 4.2. Future Research

Given the promising results observed with the HA/trehalose eyedrops, future research should focus on addressing the limitations of the current study. These studies should aim to increase the sample size and include patients with varying degrees of DED severity. Furthermore, extending the treatment period and follow-up time will provide a more comprehensive understanding of the long-term effects of this treatment.

To ensure greater precision and objectivity, future studies should strive to implement improved methods for measuring the TBUT and the NIBUT. In this regard, the utilization of advanced imaging non-invasive technologies and ocular surface analyzers could help.

Lastly, future investigations should consider the development of a standardized dosage system to control the exact amount of the dose administered to each patient. This could aid in improving the repeatability of the treatment and in reducing the variability of patient responses.

### 4.3. Clinical Application

The outcomes of this study have several potential applications under clinical settings. Clinicians can use the evidence provided here to inform their treatment choices for patients with DED. These findings support the use of HA/trehalose eyedrops, even in cases where traditional treatments have not yielded satisfactory results.

The fact that significant improvements were observed with just two daily applications suggests that this treatment may be more convenient and less burdensome for patients compared to formulations requiring more frequent instillations. This could lead towards a better compliance, enhancing the overall effectiveness of the treatment.

Moreover, the observation that combining carmellose with another wetting agent can be more effective suggests a possible new avenue for DED treatment, opening doors for further research into combination therapies.

### 4.4. Comparative Effectiveness Research

Building on the results of this study, future investigations could also focus on comparative effectiveness research. This involves comparing the benefits and harms of different interventions and strategies to prevent, diagnose, treat, and monitor health conditions. In the context of DED, this could involve comparing HA/trehalose with other popular treatments, such as lipid-based eyedrops, or with combinations of various wetting agents.

This approach could help identify the most effective and cost-efficient treatment for DED, thereby facilitating evidence-based decision making in clinical settings. Moreover, such research could provide insight into whether certain subgroups of patients respond better to specific treatments, enabling a more personalized approach to DED management.

## 5. Conclusions

In conclusion, the use of HA/trehalose eye drops, comprising 0.15% *w/v* sodium hyaluronate and 3% *w/v* trehalose, proved efficacious in reducing the signs and symptoms of the DED patients studied. This treatment was notably successful in enhancing tear stability and quality. Moreover, a regimen of two daily applications was sufficient to bring about improvement in ocular surface signs and decrease symptoms in the subjective score reported by the patients that were studied.

## Figures and Tables

**Figure 1 jcm-12-06647-f001:**
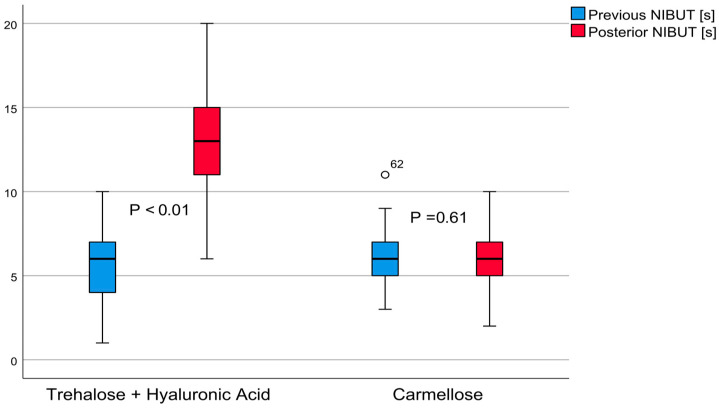
The non-invasive breakup time (NIBUT): comparative boxplots for patients treated with trehalose and hyaluronic acid eyedrops and those treated with carmellose eyedrops.

**Figure 2 jcm-12-06647-f002:**
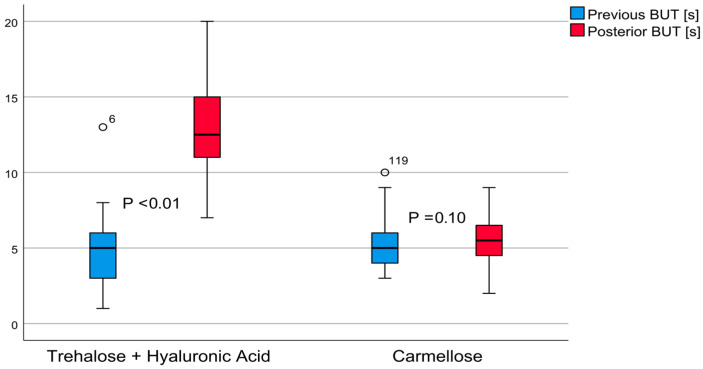
The invasive tear breakup time (TBUT): comparative boxplots for patients treated with trehalose and hyaluronic acid eyedrops and those treated with carmellose eyedrops.

**Figure 3 jcm-12-06647-f003:**
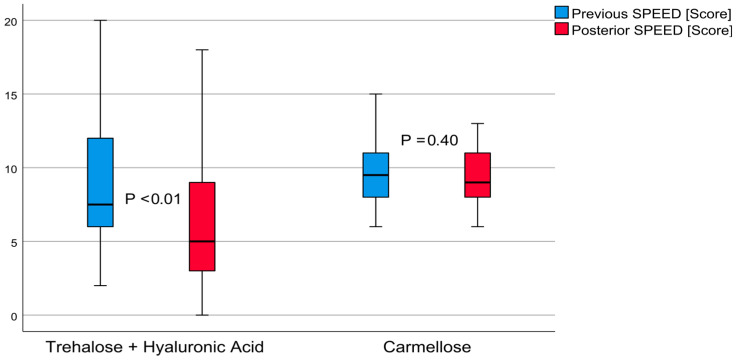
Standard patient evaluation of eye dryness (SPEED): comparative boxplots for patients treated with trehalose and hyaluronic acid eyedrops and those treated with carmellose eyedrops.

## Data Availability

The data presented in this study are available on request from the corresponding author. The data are not publicly available due to them containing information that could compromise the privacy of research participants.

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
