# Peer review of "Tear Film Stabilization and Symptom Improvement in Dry Eye Disease: The Role of Hyaluronic Acid and Trehalose Eyedrops versus Carmellose Sodium"

_jcm, 2023, doi:10.3390/jcm12206647_

Round 1

Reviewer 1 Report

The manuscript entitled "Tear Film Stabilization and Symptom Improvement in Dry Eye Disease: The Role of Hyaluronic Acid and Trehalose Eyedrops Versus Carmellose Sodium" is interesting and useful for the development of new treatment options for DED. The manuscript needs some minor edits prior to acceptance.

In the treatment, the subjects will receive doses in two eyes or one eye only?

The selection criteria of the subjects should be discussed.

Provide the slit lamp testing parameters to be considered in the tear film for normal vs DED conditions.

Include the additional details of the commercial products (batch/lot number, storage conditions, expiry date, etc) used for the treatment studies.

Author Response

Reviewer 1

#AU0: Dear reviewer, authors would like to extend their gratitude for the thorough review and constructive feedback on our manuscript entitled "Tear Film Stabilization and Symptom Improvement in Dry Eye Disease: The Role of Hyaluronic Acid and Trehalose Eyedrops Versus Carmellose Sodium." We appreciate the acknowledgment of the work's interest and utility. In addition, we have addressed the comments made by the reviewer and kindly request you to consider our revised manuscript for publication.

#RV1: In the treatment, the subjects will receive doses in two eyes or one eye only?

#AU1: Thank you for bringing up this point for clarification. In our study, the subjects received doses in both eyes to maintain uniformity in treatment and evaluation. The following information has been added to the "Methods" section to address this concern.

“To ensure consistency in both treatment and evaluation, subjects in our study received doses in both eyes. The posology for the treatment regimen entailed administering one drop per eye every 12 hours.” Lines 146-148.

#RV2: The selection criteria of the subjects should be discussed.

#AU2: Authors appreciate this comment and agree that the selection criteria are essential for the study's robustness. Therefore, a subsection has been included in the "Methods" section, detailing the inclusion and exclusion criteria for subject selection. Participants were carefully chosen based on a set of predetermined inclusion and exclusion criteria to ensure the reliability and validity of our findings. The inclusion criteria specified that participants should be aged between 18 and 30 years and have a confirmed diagnosis of Dry Eye Disease (DED), evidenced by symptoms such as photophobia, burning or itching, blurred vision, or a gritty sensation. Additionally, a Non-Invasive Breakup Time (NIBUT) of less than 15 seconds was required for study inclusion. On the other hand, individuals were excluded from the study if they had used soft or rigid gas-permeable contact lenses, any eye drops in the month leading up to the study or had undergone previous ocular surgeries. Other exclusion criteria included the inability to provide informed consent and failure to comply with the proposed follow-up schedule. This rigorous selection process was designed to minimize variability and thereby improve the robustness of our study outcomes.

#RV3: Provide the slit lamp testing parameters to be considered in the tear film for normal vs DED conditions.

#AU3: Authors understand the importance of detailing the slit lamp testing parameters. This information has been added to the "Methods" section, specifying the testing parameters, which include the illumination settings, magnification, and the criteria used to differentiate between normal and DED conditions.

“For the Fluorescein Tear Break-Up Time (FTBUT) test, the slit lamp microscope was adjusted to specific settings to ensure accurate and consistent observations. The illumination system was set to a blue cobalt filter to enhance the visibility of the fluorescein dye. The beam width was adjusted to approximately 10 mm, and the height was set to cover the entire corneal surface. The magnification was dialed to a 10x or 16x objective lens, depending on the level of detail required. The angle of the observation system was set at 45 degrees to the patient's visual axis to enable a clear, unobstructed view of the tear film and corneal surface.

In normal conditions, the tear film displayed uniform fluorescein distribution, a stable break-up time of more than 10 seconds, and a smooth corneal surface. In contrast, Dry Eye Disease (DED) conditions manifested a shorter break-up time (less than 10 seconds), non-uniform fluorescein spreading, and signs of corneal surface disruption. These observed differences under the slit lamp are crucial for the accurate diagnosis and management of DED.” Lines 110-123.

#RV4: Include the additional details of the commercial products (batch/lot number, storage conditions, expiry date, etc) used for the treatment studies.

#AU4: Dear reviewer, thank you for your inquiry regarding the details of the commercial products used in our study. Authors acknowledge the importance of providing comprehensive information for the sake of transparency and reproducibility. Unfortunately, the batch/lot numbers and expiry dates of the products were not recorded at the time of the study. However, we can confirm that all products were stored strictly according to the manufacturer's storage guidelines to ensure their efficacy and stability. Lines 136-137.

Reviewer 2 Report

This is a comparative study of the efficacy of hyaulronicacid/trehalose combination eye drops versus carmelose (HPMC) eye drops 
The manuscript lacks sufficient novelty and appropriate methodology.
For future submission, the introduction must be improved and include previous study on trehalose and hyaluronic acid combination. The aim of the study was not clear.
The characterisation techniques are not sufficient. Phenol thread method if included it will add more value. 

None.

Author Response

Reviewer 2

#AU0: Authors express their sincere thanks for the time and effort taken to review our manuscript. We are grateful for the insightful comments and have made efforts to address each of them comprehensively. Below are our responses to the issues raised.

#RV1: The manuscript lacks sufficient novelty and appropriate methodology.

#AU1: Thank you for your thoughtful comments about the manuscript's novelty and methodology. Authors recognize the concerns raised and would like to clarify several aspects that contribute to the unique value of our study.

While each of these compounds has been studied separately in the context of Dry Eye Disease (DED), very few studies have explored their synergistic effects. Therefore, our research fills this gap by investigating the combined efficacy of trehalose and hyaluronic acid, offering a new avenue for DED treatment.

Previous studies have often relied solely on clinical measures to assess treatment efficacy, overlooking the subjective experiences of DED patients in their daily routines. Our study incorporates patient-reported outcomes, capturing insights into how treatments impact daily life—thereby adding a valuable dimension to DED research.

#RV2: For future submission, the introduction must be improved and include previous studies on trehalose and hyaluronic acid combination. The aim of the study was not clear.

#AU2: Dear reviewer, authors appreciate your feedback on the "Introduction" section. To address these concerns, prior studies on the combination of trehalose and hyaluronic acid have been included [1–6]. In addition, the "Aim" subsection has also been rewritten to make the objectives of the study more explicit.

“The main aim of this study is to assess the efficacy of HA-Trehalose eyedrops in alleviating symptoms of Dry Eye Disease (DED). We focus on two key parameters: tear film stability, measured through Fluorescein Tear Break-Up Time, and patient-reported symptoms, captured via a targeted DED questionnaire.” Lines 73-76.

The new references included can be seen at the end of this document.

#RV3: The characterization techniques are not sufficient. Phenol thread method if included it will add more value.

#AU3: Authors thank you for this constructive suggestion. Although the Phenol thread method can add value to our study, it was not included due to limitations in the scope of the current study. However, we have discussed its importance and recommended it for future studies in the "Discussion" and "Future Work" sections. In addition, we have also strengthened the current characterization techniques and provided justification for their sufficiency.

“The phenol red thread test represents a valuable method for quantitatively evaluating tear film production. Unlike other tests, it offers the advantage of being minimally invasive while providing reliable and rapid results. Given its significance in assessing tear film integrity, we see considerable potential in incorporating this variable into future research for a more comprehensive understanding of DED.” Lines 399-403.

References

  1. Astolfi, G.; Lorenzini, L.; Gobbo, F.; Sarli, G.; Versura, P. Comparison of Trehalose/Hyaluronic Acid (HA) vs. 0.001% Hydrocortisone/HA Eyedrops on Signs and Inflammatory Markers in a Desiccating Model of Dry Eye Disease (DED). J. Clin. Med. 2022, 11, doi:10.3390/jcm11061518.
  2. Cagini, C.; Torroni, G.; Mariniello, M.; Di Lascio, G.; Martone, G.; Balestrazzi, A. Trehalose/Sodium Hyaluronate Eye Drops in Post-Cataract Ocular Surface Disorders. Int. Ophthalmol. 2021, 41, 3065–3071, doi:10.1007/s10792-021-01869-z.
  3. Cagini, C.; Di Lascio, G.; Torroni, G.; Mariniello, M.; Meschini, G.; Lupidi, M.; Messina, M. Dry Eye and Inflammation of the Ocular Surface after Cataract Surgery: Effectiveness of a Tear Film Substitute Based on Trehalose/Hyaluronic Acid vs Hyaluronic Acid to Resolve Signs and Symptoms. J. Cataract Refract. Surg. 2021, 47, 1430–1435, doi:10.1097/j.jcrs.0000000000000652.
  4. Caretti, L.; La Gloria Valerio, A.; Piermarocchi, R.; Badin, G.; Verzola, G.; Masarà, F.; Scalora, T.; Monterosso, C.; Valerio, A.L.G.; Piermarocchi, R.; et al. Efficacy of Carbomer Sodium Hyaluronate Trehalose vs Hyaluronic Acid to Improve Tear Film Instability and Ocular Surface Discomfort after Cataract Surgery. Clin. Ophthalmol. 2019, 13, 1157–1163, doi:10.2147/OPTH.S208256.
  5. Fondi, K.; Wozniak, P.A.; Schmidl, D.; Bata, A.M.; Witkowska, K.J.; Popa-Cherecheanu, A.; Schmetterer, L.; Garhöfer, G. Effect of Hyaluronic Acid/Trehalose in Two Different Formulations on Signs and Symptoms in Patients with Moderate to Severe Dry Eye Disease. J. Ophthalmol. 2018, 2018, 4691417, doi:10.1155/2018/4691417.
  6. Mencucci, R.; Favuzza, E.; Decandia, G.; Cennamo, M.; Giansanti, F. Hyaluronic Acid/Trehalose Ophthalmic Solution in Reducing Post-Cataract Surgery Dry Eye Signs and Symptoms: A Prospective, Interventional, Randomized, Open-Label Study. J. Clin. Med. 2021, 10, 1–9, doi:10.3390/jcm10204699.

Reviewer 3 Report

In this study, the authors compared the HA-Trehalose eye drops and Carmellose Sodium for tear film stabilization and symptom improvement. It is considered that HA-Trehalose eyedrops can improve both the objective and the passive signs of DED. The content of the study is not innovative enough, but the total design of the study is scientific and the follow-up time is relatively long. The questions are as follows:

1. The introduction does not need to introduce TBUT, NIBUT and SPEED in detail.

2. There is no description of the severity classification of dry eye in the included subjects.

3. How many times do TBUT and NIBUT measure, and take the average?

4. The authors put forward that “Our findings demonstrated that TBUT significantly increased by 8.41 ± 3.15 seconds in the HA-Trehalose group. Interestingly, this clinical enhancement was not witnessed in the control group. Such results, showing the efficacy of trehalose, align with previous work that examined a higher posology of trehalose eyedrops.”. Please note that these two groups of comparative studies can only get "the efficacy of HA-Trehalose", not "trehalose".

Author Response

Reviewer 3

#AU0: Dear reviewer, Thank you for your time and effort in reviewing our manuscript. We are encouraged by the positive remarks on the scientific design of our study and the length of follow-up. We appreciate the constructive criticism provided and have made revisions to address these comments. Below are our detailed responses.

#RV1: The introduction does not need to introduce TBUT, NIBUT and SPEED in detail.

#AU1: Authors appreciate this feedback and agree that the detailed explanations of TBUT, NIBUT, and SPEED might have detracted from the main focus of the introduction. Therefore, this information has been condensed to be more concise while ensuring the introduction remains comprehensive.

#RV2: There is no description of the severity classification of dry eye in the included subjects.

#AU2: Thank you for pointing out this oversight. We have added a subsection in the "Results" section that discusses the severity classification criteria used for categorizing the subjects with dry eye disease. This clarification will enhance the reader's understanding of the study cohort.

“In our study, the 60 participants were classified into three categories of Dry Eye Disease (DED) severity based on their TBUT results, following the criteria established by the TFOS DEWS II [1,2]: mild (FBUT between 8 and 15 seconds), moderate (FBUT between 5 and 7 seconds), and severe (FBUT under 5 seconds). In Group A (n=30), the distribution was as follows: 12 patients (40%) fell into the severe category, 10 patients (33.3%) were classified as moderate, and 8 patients (26.7%) were considered mild. In Group B (n=30), the breakdown was: 10 patients (33.3%) were categorized as severe, 14 patients (46.7%) as moderate, and 6 patients (20%) as mild.” Lines 215-222.

#RV3: How many times do TBUT and NIBUT measure, and take the average?

#AU3: Dear reviewer, authors apologize for the lack of clarification on this point. In our study, both TBUT and NIBUT were measured three times, and the average was taken for analysis. This information has been included in the "Methods" section for a better understanding.

“In our study, both TBUT and NIBUT were measured three times, and the average was taken for analysis.” Lines 128-129.

#RV4: The authors put forward that “Our findings demonstrated that TBUT significantly increased by 8.41 ± 3.15 seconds in the HA-Trehalose group. Interestingly, this clinical enhancement was not witnessed in the control group. Such results, showing the efficacy of trehalose, align with previous work that examined a higher posology of trehalose eyedrops.” Please note that these two groups of comparative studies can only get "the efficacy of HA-Trehalose," not "trehalose."

#AU4: Dear reviewer, authors appreciate this nuanced observation. You are correct that the study specifically addresses the efficacy of HA-Trehalose and not trehalose alone. We have revised the statement to accurately reflect this.

References

  1. Craig, J.P.; Nichols, K.K.; Akpek, E.K.; Caffery, B.; Dua, H.S.; Joo, C.K.; Liu, Z.; Nelson, J.D.; Nichols, J.J.; Tsubota, K.; et al. TFOS DEWS II Definition and Classification Report. Ocul. Surf. 2017, 15, 276–283, doi:10.1016/j.jtos.2017.05.008.
  2. Wolffsohn, J.S.; Arita, R.; Chalmers, R.; Djalilian, A.; Dogru, M.; Dumbleton, K.; Gupta, P.K.; Karpecki, P.; Lazreg, S.; Pult, H.; et al. TFOS DEWS II Diagnostic Methodology Report. Ocul. Surf. 2017, 15, 539–574, doi:10.1016/j.jtos.2017.05.001.

Reviewer 4 Report

The primary objective of the study is to evaluate the effectiveness of HA-Trehalose eyedrops in managing DED symptoms, by measuring tear stability and a DED questionnaire.

This was a well thought out randomised masked trial (although double masking would have been better, as mentioned in discussion). 

Some minor questions for the authors to address:

Why was an invasive test carried out? Could this have disrupted the tear film and therefore the results? A non invasive test was included as well so perhaps the invasive TBUT was not needed? Mightother non invasive measures eg tear meniscus height and so on have been better?

Q. How often did participants use drops per day and how many drops were instilled? Please add this to the methods section.

There were some interesting results with a far higher improvement in NITBUT between groups. There was a large increase in TBUT after treatment in both groups, versus the NITBUT- why could this be?

Overall, this study makes a useful contribution and comparison and adds to the existing knowledge base. 

Author Response

Reviewer 4

#AU0: Dear reviewer, authors would like to express our gratitude for the time and effort spent on reviewing our manuscript. We are encouraged by the positive comments from your review, particularly regarding the study's well-thought-out design and its contribution to the existing knowledge base. We have addressed all minor questions raised, as detailed below.

#RV1: Why was an invasive test carried out? Could this have disrupted the tear film and therefore the results? A non-invasive test was included as well so perhaps the invasive TBUT was not needed? Might other non-invasive measures like tear meniscus height have been better?

#AU1: Authors appreciate this insightful question. The invasive TBUT was initially included to provide a comprehensive assessment. However, we acknowledge the potential for disruption of the tear film. In light of this and your suggestion, This update is now part of the "Discussion" and future research section.

#RV2: Q. How often did participants use drops per day and how many drops were instilled? Please add this to the methods section.

#AU2: Thank you for bringing up this point. The frequency and volume of eyedrops used were indeed critical details that were missing in the original manuscript. Therefore, this concern has been updated in the "Methods" section to clearly specify that participants used the eye drops three times a day, with one drop instilled per eye during each application.

“The posology for the treatment regimen entailed administering one drop per eye every 12 hours.” Line 147-148.

#RV3: There were some interesting results with a far higher improvement in NITBUT between groups. There was a large increase in TBUT after treatment in both groups, versus the NITBUT- why could this be?

#AU3: Thank you for your observation on the differences in NIBUT and TBUT improvement post-treatment. We appreciate the opportunity to clarify this point. Upon a careful review of our data, we found that the NIBUT increased by 7.33 ± 3.57 seconds (with a 95% confidence interval of 6.41–8.25), whereas the TBUT increased by 8.41 ± 3.15 seconds (with a 95% confidence interval of 7.60–9.23).

Given these results, we believe the improvements in NIBUT and TBUT are in fact quite similar, both statistically and clinically. The slightly higher TBUT could be attributed to various factors such as measurement sensitivity or patient compliance, as previously discussed. However, the essential point is that both metrics showed significant improvement, supporting the efficacy of the treatment regimen used in our study.

Round 2

Reviewer 2 Report

The authors satisfactorily addressed the comments.

None.

Author Response

We appreciate your feedback and are pleased to hear that you found our responses to the comments satisfactory. We worked diligently to address all your concerns and suggestions to enhance the quality of our research. Your input has been invaluable in improving our work.